# Physical and Numerical Modeling of the Slag Splashing Process

**DOI:** 10.3390/ma14092289

**Published:** 2021-04-28

**Authors:** Viktor Sinelnikov, Michał Szucki, Tomasz Merder, Jacek Pieprzyca, Dorota Kalisz

**Affiliations:** 1LLC Technical University Metinvest Polytechnic, 71a Sechenova str., 87524 Mariupol, Ukraine; victoriano090691@gmail.com; 2Foundry Institute, Technische Universität Bergakademie Freiberg, Bernhard-von-Cotta-Str. 4, 09599 Freiberg, Germany; Michal.Szucki@gi.tu-freiberg.de; 3Faculty of Materials Engineering, Silesian University of Technology, Krasinskiego 8, 40-019 Katowice, Poland; tomasz.merder@polsl.pl (T.M.); jacek.pieprzyca@polsl.pl (J.P.); 4Faculty of Foundry Engineering, AGH-University of Science and Technology, al. A. Mickiewicza 30, 30-059 Krakow, Poland

**Keywords:** steel, oxygen-converter process, slag splashing, numerical modeling, physical modeling

## Abstract

The influence of technological factors on the process of slag splashing was analyzed in the paper. The problems were solved in several stages using our own and commercial calculation programs and laboratory tests. Based on the performed calculations and simulations, factors affecting the slag splashing were determined. It was observed that the high efficiency of the process can be achieved by optimizing numerous technological parameters, e.g., flow parameters, pressure, and temperature of the nitrogen stream, height and angle of the lance position, as well as slag height into which the gas stream enters and MgO consumption. In addition, the chemical and mineralogical composition of the slag and its physicochemical parameters should be also considered. The obtained results of numerical simulations of slag splashing in the oxygen converter coincide with the results of experiments carried out using the physical model of oxygen converter. This means that the simulations well represent the real course of the slag splashing process for the studied variants.

## 1. Introduction

A high cost of maintenance and repair of furnace refractory lining necessitates undertaking suitable measures to prevent its premature wearing or to fix the existing damage. A solution to this is the use of the slag splashing technology, in which the waste material is a by-product of the steel melting process in an oxygen converter. After the steel is melted in the oxygen converter and poured into the ladle, a certain amount of molten slag is left at the bottom of the furnace. This slag is modified with additives that increase its ability to adhere to the refractory lining of the converter. Then, the lance is lowered and a stream of nitrogen or a nitrogen-MgO powder mixture is blown into the liquid slag, resulting in its splashing on the walls of the refractory lining. The high-pressure gas blowing is carried out for 2 to 4 min, and the process results in the formation of a protective slag layer on the surface of the spent refractories, which provides better protection and maintenance of the refractory. The excess slag is then poured out of the converter to prevent clogging of the nozzles of the gas permeable fittings in the bottom of the converter (Figure 1). The following technological and design parameters influence the thickness and distribution of the applied slag layer: stream characteristics (flow rate, lance height and angle, number and dimensions of nozzles), slag physicochemical properties (viscosity, density, etc.), and system geometry. The process involves splashing liquid slag onto the surface of the refractory lining using a nitrogen stream or a mixture of gas and refractory powder. The amount of slag splashed by blowing nitrogen (N_2_) increases with the increasing gas flow rate as confirmed by simulation results of the slag splashing process on the water model [1,2,3], during laboratory simulation tests [4], and analysis of experimental data [5].

The splashing efficiency depends on the position of the lance, size and shape of the converter, and its operating conditions. By increasing the number of nozzles in the slag splash lance, the following effects can be obtained [6,7,8]: less slag is splashed in the central area, and the splashing is more uniform. The effect of slag splashing is permanent, determined by hydrodynamic conditions and surface (capillary) phenomena. However, to obtain maximum efficiency, it is necessary to solve many issues associated with establishing technological and design parameters [6,7,8,9,10]. When developing mathematical models of gas flow in nozzles, attention has to be paid to the specific character of technological conditions of splashing. During “slag splashing,” it would be necessary to maintain constant pressure in front of the nozzles of one and the same size. However, technically, this cannot be carried out because the pressure of the gas stream decreases from the level of ~1.7 MPa at the beginning of slag splashing to 1.1 MPa at the end; nevertheless, the flow still remains in the supersonic velocity range [9]. The nature of the supersonic stream entering the converter is well known and described in the literature. When the pressure in the lance differs from the pressure in the converter volume, the gas stream exiting the nozzle has a shock wave character [10].
*n* = *p_a_*/*p_g_* > 1 or < 1(1)
where *p_a_* is pressure on nozzle cross-section; *p_g_* is pressure in converter body and *n* is the nonisobaricity parameter.

This means that for an equal consumption of air introduced into the nozzles, the structure of the streams leaving the nozzles differs significantly. The higher is the value of parameter *n* different from 1, the higher is the temperature in the converter, and the bigger is the stream energy loss behind the nozzle. Therefore, the entropy change can be determined from the following formula:(2)$$\Delta S=\frac{R}{k-1}\ln\left[\frac{p_{2}}{p_{1}}{\left(\frac{\rho_{1}}{\rho_{2}}\right)}^{k}\right]$$
where *k*—nitrogen adiabatic, *R*—universal gas constant.

Energy loss is expressed as follows:(3)$$\Pi =\rho_{n}V_{n}T_{atm}\Delta S$$
where *ρ**_n_* is the density (normal conditions), *V_n_* is volumetric flow under standard conditions, *T_atm_* is the temperature in converter volume, and Δ*S* is Entropy.

Accordingly, the change of the stream rate equals to
(4)$$w_{1}/w_{2}=l_{1}/l_{2}=1.48/0.68=2.2$$

Authors of this paper present complex results of tests performed with the computer program “Slag Splashing” on the conditions of carrier gas flow in the nozzle. The effect of parameters such as gas jet power, carrier gas temperature, and nozzle nitrogen consumption on thermogasodynamic flow parameters was analyzed. Then, the addition of a refractory mixture and its effect on the magnitude of the same parameters was considered. The influence of pressure on fluid flow was also considered for different nozzle parameters and thermal conditions (carrier gas and converter workspace). These simulations made it possible to study the influence of particular quantities characteristic of the slag splashing process on the nature of the medium flow in the nozzle. In industrial conditions the process is carried out using a lance equipped with an array of nozzles; therefore, the simulations in Flow-3D program were performed using a lance with a system of nozzles for several variants of their arrangement. This was performed to visualize the slag spattering, followed by the visual verification on a physical model (Figure 2).

## 2. Materials and Methods

Modeling studies included computer simulations of the nozzle operation with the use of a computer program (Slag Splashing Program) based on the mathematical model of gas flow and gas-powder mixture through the nozzles, modeling with the Flow-3D computer program and a physical model. Model tests in the field of CFD calculations were carried out with the Flow-3D program, version 10 [11]. This program can be also used to model two-phase flows. It employs a modified volume of fluid method, the so-called TrueVOF [12]. This method includes the following components: a scheme to determine the location of the free surface, an algorithm to track this surface as an interface moving through the computational lattice, and tools to introduce boundary conditions on the free surface [11,12,13,14,15]. Simulations in this paper were performed for a 3D system geometry. During stage 1, the results for the converter symmetry axis cross section (2D) were analyzed for three different variants of lance nozzle arrangements. In this way, the optimum process parameters of the slag spattering could be determined. Then, the results were visualized and analyzed for the thus selected model, (step 2) for a 3D system geometry. The results of these calculations were verified using a cold physical model of an oxygen converter made on a scale of 1:10. A modified Froud number was used as a dominance similarity criterion. On its basis, the dynamic and kinematic similarity was determined by the method of scales. The results of tests carried out using the cold converter model were compared with the results of 3D numerical simulations. A good convergence was obtained for the results proving their high reliability.

### 2.1. Mathematical Modeling—Slag Splashing Program

Calculations were performed using a computer Slag Splashing Program developed at the Pryazovskyi State Technical University in Mariupol, Ukraine. The influence of the following parameters on the slag splashing process was analyzed: nitrogen temperature, nitrogen consumption, powder material consumption, and stream pressure for different variants. The applied mathematical model takes into account many parameters, the most important of which are the following:

The heat capacity of nitrogen is
(5)$$c_{p_{1}}=\frac{k}{k-1}R_{1}$$
and
(6)$$c_{v_{1}}=\frac{1}{k-1}R_{1}$$

Constant *R*_12_ of carrier gas N_2_ and refractory powder (MgO) mixture is
(7)$$R_{12}=\frac{R_{1}}{1+\mu \psi }$$
where *μ* is the relation of mass charges of gas and powder coefficient) = *m*_2_/*m*_1_, *m*_2_ is mass consumption of nitrogen by a nozzle, *m*_1_ is powder consumption, and *ψ* is the parameter of speed of flow = *w*_2_/*w*_1_.

The density of the gas stream (in lance) is expressed with
(8)$$\rho_{0}=\frac{p_{0}^{/}\left(1+\mu \right)}{R_{1}T_{0}}$$

The two-phase flow index is
(9)$$ℵ=\left(k-1\right)\left(1-\phi_{1}\right)+1$$

The critical value of two-phase flow equals to
(10)$$a_{kp}=\sqrt{\frac{2ℵ}{ℵ+1}\frac{1}{1+\mu \psi }R_{1}T_{0}}$$

The real pressure beyond the nozzle is described with the equation
(11)$$p_{0}^{/}=\frac{1}{B}\frac{m_{12}\sqrt{R_{12}T_{0}}}{F_{kr}}=\frac{1}{B}\frac{\rho_{n}V_{n}(1+\mu \psi )\sqrt{R_{12}T_{0}}}{Fkp}$$
where coefficient *B* is expressed as
(12)$$B=\sqrt{ℵ{\left(\frac{2}{ℵ+1}\right)}^{\frac{ℵ+1}{ℵ-1}}}$$

The density of stream on the outlet nozzle cross-section is expressed with dependence as follows:(13)$$\rho_{1}=\rho_{0}\varepsilon \left(\lambda_{1}\right)=\frac{p_{0}^{|}\left(1+\mu \psi \right)}{R_{1}T_{0}}{\left(1-\frac{ℵ-1}{ℵ+1}\lambda_{1}^{2}\right)}^{\frac{1}{ℵ-1}}$$

On the other hand, the nitrogen temperature on the outlet nozzle cross section is expressed as
(14)$$T_{1}=\frac{p_{1}}{\rho_{1}R_{1}}=\frac{np_{g}\left(1+\mu \psi \right)}{\rho_{1}R_{1}}$$

The temperature of nitrogen after it leaves the nozzle equals to
(15)$$T_{01}=T_{1}+\frac{w_{1}^{2}\left(ℵ-1\right)\left(1+\mu \psi \right)}{2ℵR_{1}}$$
where parameter *ψ*_1_ takes the following form:(16)$$\psi_{1}=\frac{1}{{\left(M_{i}^{2}-1\right)}^{1/2}}\frac{q\left(M_{i}\right)}{q\left(M_{1}\right)}$$

Additionally, parameter *I* is
(17)$$I_{1R}=\int_{-\infty }^{\eta_{R}}\frac{\phi d\eta }{\theta +\phi \left(1-\theta \right)-\phi^{2}C_{i}^{2}}$$
(18)$$I_{2R}=\int_{-\infty }^{\eta_{R}}\frac{\phi^{2}d\eta }{\theta +\phi \left(1-\theta \right)-\phi^{2}C_{i}^{2}}$$

The nitrogen stream in the converter body at a distance *x* from the nozzle totals to
(19)$$g=\frac{2r_{\max}x}{D\sigma }C_{i}{\left(1-C_{i}^{2}\right)}^{1/2}\left(I_{1R}-I_{2R}\right)$$
$$C\;\mathrm{is}\;\mathrm{the}\;\mathrm{number}\;\mathrm{of}\;\mathrm{Crocco}=C=\sqrt{1-{\left(1-\frac{k-1}{2}M^{2}\right)}^{-1}}$$
where *M—*the number Mach at the boundary of the jet and the surrounding gas; *k—the* ratio of specific heats; $D={\left(\left(k-1\right)/2\right)}^{0.5}\cdot Ma_{1}\cdot n^{\frac{k+1}{2k}}$*—*establishes a relation between the Mach number *Ma*_1_ at the nozzle cross section and *n*, *σ =* 12 + 2.58 *Ma*; *r_max_—*is the relative maximum radius of the first cell in the mismatched supersonic jet with density discontinuities and is calculated by analogy with [16,17,18]; *x*
*= l/**r*_1_ is the distance from the nozzle cross section to the cross section that is under consideration along the jet axis (measured in nozzle diameters); *r*_1_ is the nozzle’s output radius.

The heat capacity of the stream leaving the nozzle is
(20)$$c_{p_{0}}=c_{p_{1}}{\bar{g}}_{1}+c_{2}{\bar{g}}_{2}$$
where the heat capacity of nitrogen is
(21)$$c_{p_{g}}=c_{p_{N_{2}}}=1.25$$

On the other hand, the heat capacity of slag equals to
(22)$$C_{\omega }=0.276+1.138\cdot 10^{-3}t_{\omega }$$
where *t**_ω_* is the slag temperature.

The consumption of particular components in the process was described with the following expressions:-participation of nitrogen and MgO after leaving the nozzle:
(23)$$\bar{g}=\frac{m_{1}}{m_{1}+m_{g}+m_{ż}}=\frac{1}{1+\frac{m_{g}}{m_{1}}+\frac{m_{ż}}{m_{1}}}=\frac{1}{1+g+g_{ż}}$$
-participation of gas in the converter volume:
(24)$${\bar{g}}_{g}=\frac{m_{g}}{m_{1}+m_{g}+m_{ż}}=\frac{1}{1+\frac{m_{1}}{m_{g}}+\frac{m_{ż}}{m_{g}}}=\frac{1}{1+\frac{1}{g}+\frac{g_{ż}}{g}}=\frac{1}{1+\frac{1}{g}\left(1+g_{ż}\right)}$$
-participation of slag in the converter volume:
(25)$${\bar{g}}_{ż}=\frac{m_{ż}}{m_{1}+m_{g}+m_{ż}}=\frac{1}{1+\frac{m_{1}}{m_{ż}}+\frac{m_{g}}{m_{ż}}}=\frac{1}{1+\frac{1}{g_{ż}}+\frac{g}{g_{ż}}}$$

The heat capacity of nitrogen and MgO after leaving the nozzle and slag in the converter volume was described with the relation
(26)$$\begin{array}{l} c_{\rho x}=\sum c_{i}g_{i}=c_{\rho_{1}}\bar{g}+c_{\rho_{g}}{\bar{g}}_{g}+c_{ż}{\bar{g}}_{ż}=c_{\rho_{1}}{\bar{g}}_{1}+c_{2}{\bar{g}}_{2}+c_{\rho_{g}}{\bar{g}}_{g}+c_{ż}{\bar{g}}_{ż}=\\ =c_{\rho_{1}}\frac{1}{1+\mu }+c_{2}\frac{\mu }{1+\mu }+c_{\rho_{g}}{\bar{g}}_{g}+c_{ż}{\bar{g}}_{ż} \end{array}$$

The rate of N_2_ + MgO stream, nitrogen exiting de Laval nozzle in the converter volume, and the associated slag stream are expressed as
(27)$$w_{x}=\left[\left(1-g_{ż}\psi_{ż}\right)w_{1}+\frac{p_{g}\left(n-1\right)}{\rho_{1}w_{1}}-\frac{F_{x}}{F_{1}}\frac{p_{x}-p_{g}}{\rho_{1}w_{1}}\right]\frac{1}{\left(1+g+g_{ż}\right)\beta };$$
where flow turbulence coefficient (about 1.03).

if *p_x_* = *p*_1_:(28)$$w_{x}=\left[\left(1-g_{ż}\psi_{ż}\right)w_{1}+\frac{p_{ż}\left(n-1\right)}{\rho_{1}w_{1}}\right]\frac{1}{\left(1+g+g_{ż}\right)\beta }$$

Stream temperature *T_x_* at a distance *x* from the nozzle cross section is described with the relation
(29)$$T_{x}=\frac{T_{01}+g\frac{c_{pg}}{c_{p_{0}}}T_{g}+g_{ż}\frac{c_{ż}}{c_{p_{0}}}T_{ż}-\alpha \left(1+g+g_{ż}\right)\frac{w_{x}^{2}}{2000}}{\left(1+g+g_{ż}\right)\frac{c_{px}}{c_{p_{0}}}}$$

The total stream pulse at a distance *x* from the nozzle cross-section equals to
(30)$$\begin{array}{l} i_{x}=(m_{0}+m_{g}+m_{ż})w_{x}=m_{1}\left(1+\mu \right)(1+g+g_{ż})w_{x}=\\ =\rho_{n}V_{n}\left(1+\mu \right)\left(1+g+g_{ż}\right)w_{x} \end{array}$$

The strength of the multiphase stream at a distance x from the nozzle cross section is described with the following formula:(31)$$N_{x}=\frac{\alpha (m_{1}+m_{g}+m_{ż})w_{x}^{2}}{2000}=\frac{m_{1}\left(1+\mu \psi \right)\left(1+g+g_{ż}\right)w_{x}^{2}}{2000}$$
where *α* is the kinetic energy coefficient.

### 2.2. Computer Simulation

In the first step, the effect of nitrogen stream temperature on the stream parameters was analyzed. When the temperature *t*_0_ is increased, the stream velocity also increases (Figure 3a). For the same reason, *N_x_* (stream power) in each nozzle cross-section *x* increases when the nitrogen stream is heated. When temperature *t*_0_ increases to 600 °C, temperature *t_x_* increases over three times to 1780 °C, and so does power *N_x_* = 0.61 MW [16].

In turn, the analysis of Figure 3b reveals that with the increasing nitrogen temperature *t*_0_ (here nitrogen heating temperature), the assumed mass of oxygen g decreases [17]. For example, with the increase of *t*_0_ from 30 °C to 600 °C at a distance *x* = 20 calibers, the assumed mass of oxygen *g* decreases from 0.94 to 0.4. This phenomenon is caused by the fact that with the increase of *t*_0_ the initial pressure of mixture *p*_0_ simultaneously increases, and therefore, *g* decreases. This means that with the increase of temperature *t*_0_ at a distance *x* = 30 calibers, the power of nitrogen stream *N_x_* increases from 0.19 MW to 0.62 MW.

Both the gas stream flow modeling and the results of computer simulations show that with the increase of gas temperature in the converter volume, the flow rate and the gas temperature after exiting the nozzle also grow. These conclusions confirm the influence of gas temperature *t_g_* in the converter on the effectiveness of the slag spattering process. (Figure 4a) [17]. For instance, at a distance $\bar{x}=\mathrm{20}$ calibers, at *t_g_* = 1600 °C, the ratio *T_x_*/*T*_1_ = 4.75, and *w_x_*/*w_1_* = 2.6 [17]. When the converter gas is cooled, the participation of *T_x_*/*T_1_* in the furnace is reduced from 2.5, and rate *w_x_*/*w_1_* to 1.9, which in turn results in 1.87 times lower supersonic stream power.

The increase of temperature of the gas-powder stream before the nozzle *t*_0_ results in a significant increase in the temperature of the supersonic stream at a distance $\bar{x}$ and an insignificant increase in *N_x_*/*N_1_* power. For instance, at *x* = 20 and at temperature *t*_0_ = 25 °C, the temperature of the supersonic stream equals to *t_x_* = 585 °C, and *N_x_*/*N*_1_ is 0.41. On the other hand, the increase of temperature of the gas-powder stream before the nozzle *t*_0_ causes an increase of temperature *t_x_* = 645 °C and growth of *N_x_*/*N*_1_ power to 0.48 (Figure 4b).

Nitrogen consumption *V_n_* of the nozzles significantly affects the thermogasodynamic flow parameters (Figure 5a). For example, if gas *consumption V_n_* by the nozzles increases from 180 m^3^·min^−1^ to 380 m^3^·min^−1^, then at a distance $\bar{x}=20$ calibers, the *T_x_*/*T*_1_ ratio decreases from 3.69 to 1.85 [17]. This is caused by the fact that with the growing value of *V_n_* the gas mass distributed in the converter volume lowers. Therefore, with the increase of nitrogen consumption by the nozzles, stream temperature *t_x_* lowers, as does the *T_x_*/*T*_1_ ratio. It should be also observed with the increase of nitrogen consumption *V_n_* in the same nozzle, temperature *t_x_* decreases, and the values of rates *w_x_* and *w_x_*/*w*_1_ correspondingly increase.

Mass consumption of refractory powder in relation to the mass consumption of nitrogen *µ* (i.e., refractory powder concentration) significantly affects parameters *t_x_*/*t*_1_ and rate *w_x_*/*w*_1_ at a distance $\bar{x}$ from the nozzle cross section (Figure 5b). For instance, at a distance $\bar{x}=20$ calibers, when the ratio of mass consumption of refractory powder to the mass consumption of nitrogen *µ* increases from 0 to 1.4, the value of *t_x_*/*t*_1_ lowers from 7.45 to 3.3, whereas rate ratio *w_x_*/*w*_1_ increases from 0.27 to 0.39 [17]. Figure 6 shows mass *g* and stream power *N_x_* to mass consumption of refractory powder to mass consumption of nitrogen *µ* ratio at a distance *x* from the nozzle cross section.

The results of the authors’ computer simulations on the influence of pressure were used for a number of unique nozzle dimensions (*d*_min_, *d*_kr_, *d*_a_). It follows from Figure 7 that with the increase of initial pressure *p*_0_, both static pressure *p* and gas flow rate *w*_1_ increase in an arbitrary nozzle cross section, taking into account the gas flow. For instance, in a situation when the distance from the nozzle inlet cross section is *l*_1_ = 100 mm, with the increase of pressure *p*_0_ from 0.8 to 1.8 Mpa, static pressure *p* grows from 0.18 to 0.22 MPa, and rate *w*_1_ increases from 455 m·s^−1^ to 511 m·s^−1^ [8].

### 2.3. Mathematical Modeling (Flow-3D)

During the slag splashing process, molten slag on the converter bottom is splashed to the converter sidewalls by means of a supersonic stream of gaseous nitrogen. The momentum of the nitrogen stream is transferred onto the molten slag, which causes the slag to be stirred and ejected by the action of a standing wave and high shear forces, respectively.

The modeling of such a complex process can be carried out with a mathematical model consisting of equations for fluid flow, mass balance, turbulence, and multiphase flow.

It was assumed that both slag and gas flows are incompressible. This assumption greatly simplifies the calculation process, significantly reducing the simulation time and increasing simulation stability. It should be noted that the omission of compressibility for higher velocities can lead to errors. Nonetheless, this practice is common in modeling metallurgical processes, including slag spattering. [1,2,3,4,5,6,12,13,15,16,17,18,19,20,21,22,23].

The flow of an incompressible Newtonian fluid is governed by the Navier–Stokes equations in the following form [13,14]:(32)$$\rho \left(\frac{\partial \vec{u}}{\partial t}+\vec{u}\cdot \nabla \vec{u}\right)=-\nabla p+\mu_{\mathrm{eff}}\nabla^{2}\vec{u}+\rho \vec{g}+F,$$

The above equation must be supplemented by the stream continuity equation (Newtonian conservation of mass principle) as follows:(33)$$\frac{\partial u_{j}}{\partial x_{j}}=0,$$
where *ρ* is the fluid density, $\vec{u}$ is the velocity vector, *t* is time, *p* is pressure, *μ*_eff_ is the effective fluid viscosity, $\vec{g}$ is the gravity vector, *F* represents additional forces applied or acting on the fluid, and *u_j_* and *x_j_* are the *j*-th components of the velocity vector and the coordinate system, respectively.

Turbulence was simulated by means of the classical two equations *k*–*ε* model [15] as follows:(34)$$\begin{array}{c} \rho v_{j}\frac{\partial K}{\partial x_{j}}=\frac{\partial }{\partial x_{j}}\left(\frac{\mu_{t}}{\sigma_{K}}\frac{\partial K}{\partial x_{j}}\right)+\mu_{t}\frac{\partial v_{j}}{\partial x_{j}}\left(\frac{\partial v_{j}}{\partial x_{j}}+\frac{\partial v_{j}}{\partial x_{i}}\right)-\rho \varepsilon ,,\\ \rho v_{j}\frac{\partial \varepsilon }{\partial x_{j}}=\frac{\partial }{\partial x_{j}}\left(\frac{\mu_{t}}{\sigma_{\varepsilon }}\frac{\partial \varepsilon }{\partial x_{j}}\right)+C_{1}\mu_{t}\frac{\varepsilon }{K}\frac{\partial v_{j}}{\partial x_{i}}\left(\frac{\partial v_{j}}{\partial x_{j}}+\frac{\partial v_{j}}{\partial x_{i}}\right)-C_{2}\frac{\varepsilon }{K}\rho \varepsilon , \end{array}$$
where *K* is the turbulent kinetic energy, *ε* is the dissipation rate, the model constants are *C*_1_ = 1.44, *C*_2_ = 1.92, *σ_k_* = 1.0 and *σ_ε_* = 1.3 [15], *μ*_eff_ is effective viscosity (*μ*_eff_ = *μ*_0_ + *μ_t_*), and *μ*_0_ is the laminar viscosity.

The turbulent viscosity *μ_t_* is calculated as
(35)$$\mu_{t}=\rho C_{\mu }\frac{K^{2}}{\varepsilon },$$
where *Cμ* = 0.09 and *K* and *ε* are determined from (34).

As already mentioned, the Flow-3D algorithm for phase separation boundary tracking is based on the volume of fluid. The VOF model is based on the assumption that two or more phases are not interpenetrating. Each *i*-th phase has associated a volume fraction *V_i_*, and in every control volume, the sum of the volume fractions of all phases is equal to 1. In the volume of fluid model, the tracking of the interface between the phases is calculated with the continuity equation for each phase [12] as follows:(36)$$\frac{\partial V_{i}}{\partial t}+\vec{u}\nabla V_{i}=0,$$

### 2.4. Computer Simulations Flow-3D

The assumed physical dimensions of the considered converter correspond to an actual industrial converter of 320 Mg of capacity, i.e., height 10.1 m, diameter 7.8 m. Nitrogen is introduced into the oxygen converter through a lance placed on its axis of symmetry. The dimensions of the applied lance and the nozzle parameters adopted in the analysis are described in Table 1.

The calculations in step 1 were carried out for three different design variants differing in the position and number of nozzles in the nitrogen blowing lance. The final stage 2 calculations were then carried out for the ultimate optimal process variant.

In the case of the first stage, the computational grid consisted of 250,000 computational cells, which guaranteed to obtain results in a short time. For the final calculations in the second stage, the computational grid consisted of 1,000,000 cells, which, in turn, was to ensure the quality of the results analyzed in a 3D environment. Due to the high dynamics of the fluid flow, the duration of the process mapped in the simulation was 2.5 s, at which the multiphase flow is fully developed. Calculations were performed for nonstationary conditions with an initial time step of 0.00007 s.

The physicochemical properties used for the CFD calculations are presented in Table 2.

Surface tension was also taken into account in numerical calculations, where the surface tension coefficient t was 0.35 kg·s^−2^, while the contact angle was 60°. On the other hand, the no-slip boundary condition was assumed for the walls of the system (converter and lance).

### 2.5. A Simulation of the Behavior of Slag during Gas Blowing on a Reduced Water Model

The design and construction of the physical model of the oxygen converter were realized according to the principles of similarity theory [23]. The physical model of the oxygen converter is made on a linear scale *S_L_* = 1:10 from transparent materials of PMMA type due to which visualization investigations are possible. The model is equipped with a specialized control and control apparatus. This element of the model includes a precise regulator of airflow blown through the lance and air supply system.

If the results of tests performed with the use of a physical model are to be implemented to industrial conditions, it is the geometric similarity as well as dynamic and kinematic similarity of phase flow that must be fulfilled [24,25]. The basic mathematical model used for determining the similarity criteria was the Navier–Stokes equations model. The dimensional analysis of these equations according to Buckingham’s theorem shows that the relevant similarity criteria for the studied process are Froude number (Fr), Euler number (Eu), Reynolds number (Re), and Weber number (We). Taking into account the aim of the research, the complexity of the studied process, and the difficulties in meeting all the similarity criteria at the same time, the Froud number (Fr) was set as the dominant criterion. The model used is therefore a model with incomplete compliance with the relevant similarity criteria. In the analyzed case, we have a multiphase fluid flow in the oxygen converter, and therefore, a modified criterion was used [23] as follows:(37)$${N^{\prime }}_{\mathrm{Fr}}=\frac{\rho_{g}\cdot u^{2}}{\rho_{1}\cdot g\cdot L},$$
where *ρ_g_* is gas density, *ρ*_l_ is liquid density, *g* is the acceleration of gravity, *L* is the characteristic dimension, and *u* is velocity.

To achieve the similarity of the dynamic and kinematic phase flow in the model and in the real device, the criterion number matching principle was used. When considering similarity conditions, the properties of de Laval nozzles used in the lance head were also taken into account. Their geometric and dynamic parameters in the water model were determined according to the method [26,27]. In this method, the stream pulse is determined, and Mach numbers *Ma* are accounted for as
(38)$$I=\frac{i}{\rho_{l}\cdot g\cdot L^{3}},$$
(39)$$\mathrm{Ma}=\sqrt{\frac{2\cdot \lambda^{2}}{\left(k+1\right)-\left(k-1\right)\cdot \lambda^{2}}},$$
where *I* is dimensionless stream pulse, *λ* is dimensionless velocity factor, and *k* is adiabat coefficient.

On this basis, the dynamic similarity of airflow through the oxygen converter model was determined with the scale method [28].

Detailed parameters of the experiment are presented in Table 3.

The view of the physical model of the oxygen converter and the control system is shown in Figure 8.

The experiments with the oxygen converter physical model were focused on the visualization of the slag spattering hydrodynamics in its workspace and were recorded with a video camera. They did not involve studies of slag interaction with the surface of the refractory lining. Therefore, quantities such as kinematic viscosity and surface tension were taken into account only for determining the hydrodynamic similarity.

Variant 3 (lance 0.2 m in diameter with four nozzles placed at an angle of 14 degrees and one nozzle perpendicular to the liquid slag) was considered in the physical tests. Polystyrene granules were used as a model fluid representing modified furnace slag, assuming that under certain conditions bulk materials can be treated as liquids in fluid theory. This fact was also taken into account while determining the similarity conditions of the model to the real object. Air was taken as the medium representing nitrogen.

## 3. Results and Discussion

### 3.1. Numerical Modeling—Stage 1

Figure 9 presents results of simulations (distribution of liquid phases) for variant 1 (variant with a lance equipped with one nozzle placed perpendicular to the surface of liquid slag in the oxygen converter). Colors in the figure show a volumetric distribution of liquid phase (slag) in grid cells.

The analysis of the results presented in Figure 9 reveals that after a time of 0.05 s (Figure 9a,b), a stream of flowing gas is introduced into the layer of liquid slag, but its power is insufficient to place the slag on the walls of the converter and detach the whole mass from the bottom of the converter. A similar phenomenon was observed as the slag splashing process continued (Figure 9c,d). The gas stream coming out of the nozzle had the energy to tear off about 40% of the liquid slag in the oxygen converter and transport it toward the walls (Figure 9e). It was observed (Figure 9f) that some slag was stuck to the right wall of the converter at a height of 6–6.5 m.

Figure 8 presents the simulation results (distribution of phases) for variant 2 (variant with a lance equipped with four nozzles placed at an angle).

It follows from the results of simulation performed for the time of 0.05 s (Figure 10a), that the slag scattering is more uniform and effective than in the previously presented calculations (Figure 4a). The energy of gas streams coming out of the nozzles tears off the slag from the bottom of the converter already in the initial stage and moves it toward the walls. It is also noted (Figure 10b—time 0.45 s) that about 85% of the slag volume breaks away from the bottom and moves toward the oxygen converter walls. However, after time 0.65 s (Figure 10c) a negative phenomenon is observed. The power of the gas stream from the lance with four nozzles is not sufficient to disperse the slag to the upper parts of the refractory lining of the oxygen converter. The slag moves toward the lance, and after 0.85 s (Figure 10d), it falls back to the bottom of the device.

Figure 11 shows the simulation results (distribution of phases) for variant 3 (variant with a lance equipped with a system with five nozzles—four nozzles placed at an angle, one nozzle perpendicular to the liquid slag surface) in the oxygen converter.

The analysis of the CFD simulation results (Figure 11) reveals that after a time of 0.1 s, the spray is very uniform, while the slag is lifted and moved toward the oxygen converter walls (Figure 9a). This is expected from the point of view of the effectiveness of the further spattering process. At this stage, the slag is initially stirred up by the interaction between the liquid phase (slag) and the gas phase (nitrogen stream). At this point, the gas phase energy should be sufficient to initiate the slag movement toward the converter walls and sufficiently dispersed to make the excitation relatively uniform over the whole surface of the converter. Analyzing further results of the slag splashing process (Figure 11b), i.e., after the time of 0.45 s, the slag mass inside the oxygen converter is almost completely removed. The further course of the process only confirms that the parameters and lance design were chosen correctly (Figure 11c). Presented calculation results show that part of the deposited slag falls back to the bottom of the converter but subsequent calculation steps confirm that the power of the outgoing gas stream is sufficient to re-lift the falling part of the slag to a height of almost 9.0 m (Figure 11d).

### 3.2. Verification—Stage 2

The preliminary CFD calculations show that variant 3 (variant with a lance equipped with a five-nozzle system—four nozzles placed at an angle, one nozzle placed perpendicular to the liquid slag surface) was the best option. Therefore, complementary CFD calculations for a 3D oxygen converter geometry were performed and followed by experiments on a physical model to verify the results. A summary of these hybrid test results is presented in Figure 12. The scale in the figure helps to interpret the plot and represents a dimensionless height of the models.

The verification relied on comparing the results of visualization of the slag spattering in the working space of the oxygen converter during the blowing phase, obtained from numerical calculations and physical modeling. The comparison was of qualitative nature. In order to clearly visualize this process, the scope of the distribution of phases was narrowed in the CFD visualization—the blue color illustrates the phase of the liquid slag in the oxygen converter.

Time parameters of slag blowing in the working space of oxygen converter, determined by CFD modeling methods, represent the real object state. The physical modeling was carried out for the model built at a scale of 1:10. Therefore, the obtained values of time parameters in studies using the physical model had to be converted to the real values. This was carried out using the scale method. The time scale, determined on the basis of the Froude similarity criterion, can be written in the form
(40)$$S_{t}={\sqrt{S}}_{L}$$

Thus, it was possible to determine the blowing condition of the slag modeling medium at the required moment of its duration under both in CFD visualization and in the water model. Due to this the results of process visualization carried out with both methods could be directly compared.

In the first period, i.e., from the start of the experiment to 0.45 s, the slag was moved evenly under the influence of the blown gas and its particles were directed toward the converter walls. After 1 s, the splashed slag reaches the level of the pivots and is carried away to the whole surface of the refractory lining of the converter. After 2 s, the splashed slag reaches the required circulation in the space of the converter.

The analysis of the results (Figure 12) shows a good match between the visualization images obtained both in the variant making use of a physical model of the converter and in the CFD modeling. This fact is a kind of verification of the correctness of the adopted mathematical procedures and suggests the reliability of the results in relation to the real object. The performed 3D visualization also correlates with the results of numerical simulations 2D.

## 4. Conclusions

The analyzed model creates bases for numerical modeling of the nozzle operation in a wide range of carrier gas flow, also when the refractory powder is added (*μ* = 0–30 kg nitrogen/kg powder). The numerical calculations with the slag splashing program show that the use of heating of the carrier medium (nitrogen) is advantageous for the slag splashing effect as then an additional temperature effect takes place affecting the slag heating and hence also the physicochemical parameters. The more durable is the effectiveness of pneumatic slag splashing on the hot lining of the oxygen converter, the higher is the power of the splashing gas flow rate. At the same time, simulation results show that the slag spattering affects the interaction of supersonic gas streams with liquid slag; even a small amount of slag added to the stream significantly inhibits its velocity. It has also been observed that preheating carrier gas, e.g., by heat recovery from the process, lowers the cost of the operation as the nitrogen consumption is reduced without changing the stream power. With the applied calculation procedure, the nozzle shape can be so selected to ensure the optimum flow of the gas stream or gas-powder mixture for any gas pressure variant. Consequently, with the selected process parameters the power of the stream flowing into the slag can be increased for the same gas or gas-powder mixture consumption. Modeling with the program creates an opportunity to improve and develop the slag splashing technology further by introducing changes in the nozzle design and using heat recovery from the process. Simulations with the Flow-3D program show that the most effective slag distribution on the converter walls was obtained for the lance with a diameter of 0.2 m and four nozzles placed at an angle of 14° with an additional, centrally placed nozzle, perpendicular to the liquid slag surface. A similar design is currently in operation at the “ArcelorMittal Kryvyi Rih” steel plant in Ukraine and at “NLMK” in Russia.

An effective and frequently used method of verifying the results of numerical calculations is the physical modeling of the examined process. In this way, the spectrum of the studied phenomena can be extended, and the reliability of the obtained results increased. The results of numerical simulations of slag spattering in an oxygen converter presented in this paper coincide with the results of experiments carried out using the physical model of the oxygen converter. This verification suggests that the calculations are correct, and the simulation well represents the real course of the slag splashing process for the investigated variants.

## Figures and Tables

**Figure 1 materials-14-02289-f001:**
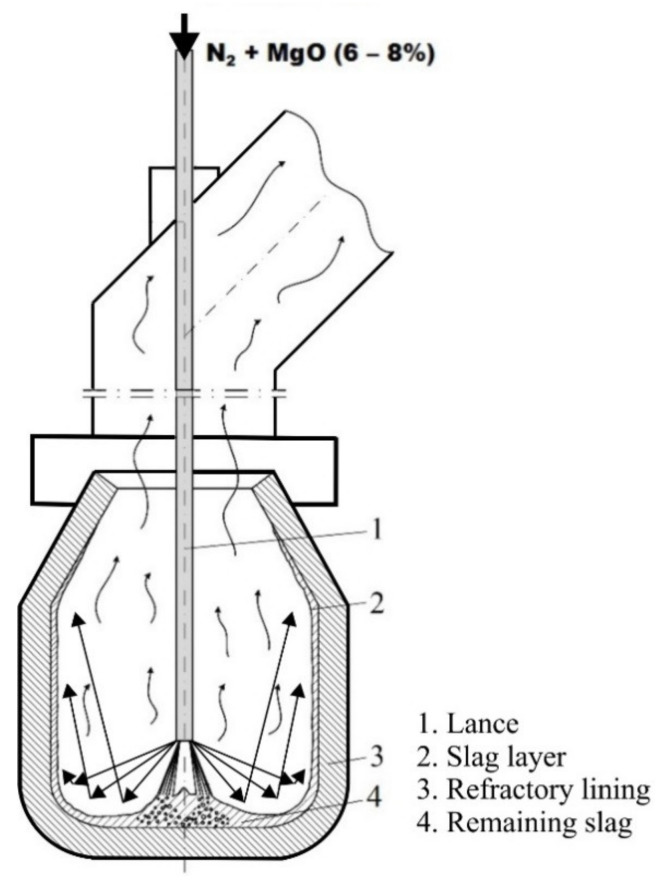
Scheme of slag splashing in the oxygen converter.

**Figure 2 materials-14-02289-f002:**
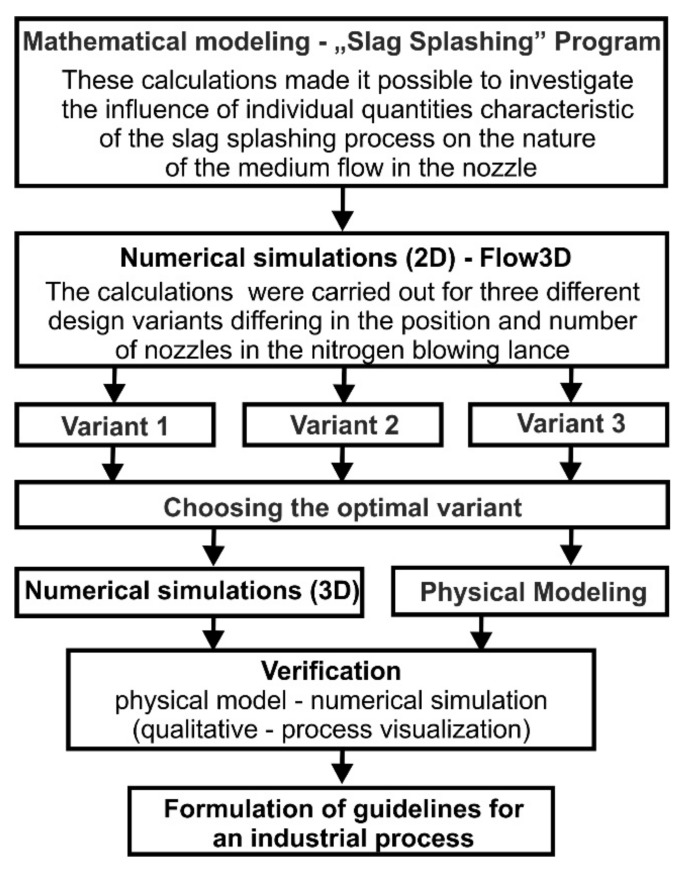
Scheme research plan.

**Figure 3 materials-14-02289-f003:**
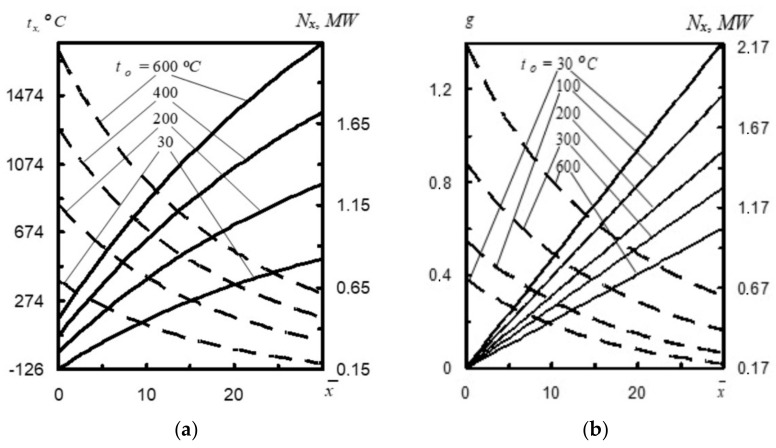
(**a**) Effect of temperature *t_x_* (––) and power *N_x_* (- ) depending on the change of nitrogen temperature *t*_0_, at a different distance *x* from the nozzle cross-section [1] and (**b**) change of assumed stream mass *g* (––) and its power *N_x_* (- ) depending on nitrogen temperature *t*_0_ at a different distance x from the nozzle cross section [16,17].

**Figure 4 materials-14-02289-f004:**
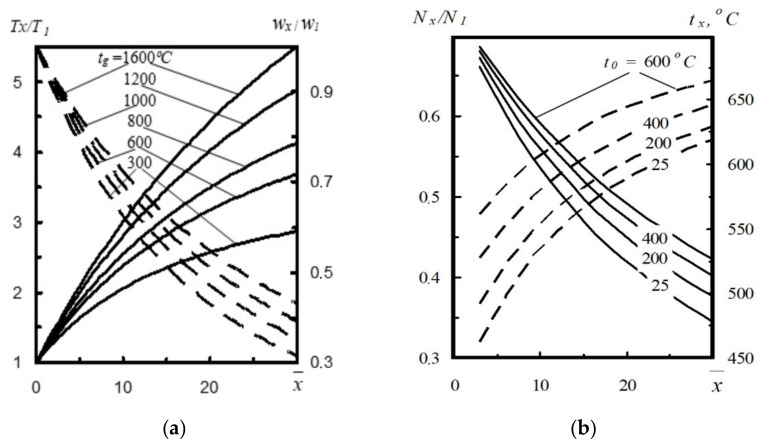
(**a**) Change of *T_x_*/*T*_1_ (––) and rate *w_x_*/*w*_1_ (- ), depending on the change of temperature in the converter body *t_g_* at a different distance x from the nozzle [2] and (**b**) influence of nitrogen temperature *t*_0_ on temperature *t_x_* (––) and ratio *N_x_*/*N*_1_ (- ) at a different distance *x* from the nozzle [17].

**Figure 5 materials-14-02289-f005:**
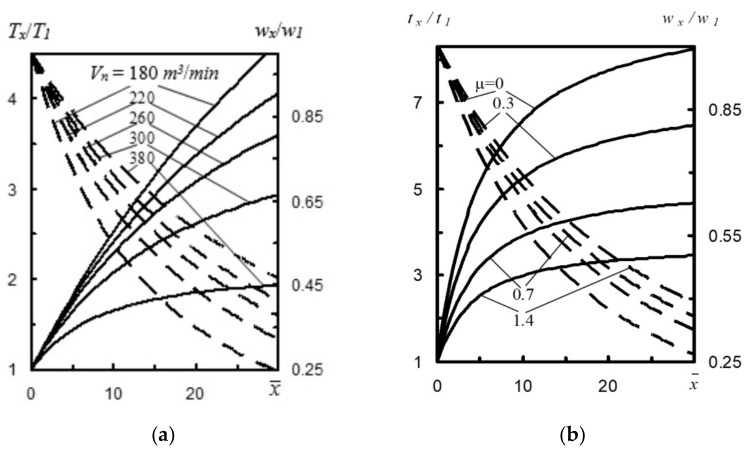
(**a**) Nitrogen consumption by nozzles vs. change of temperatures *T_x_*/*T*_1_ (––) and rate ratio *w_x_*/*w*_1_ (- ) at a different distance *x* from the nozzle cross section [17] and (**b**) mass consumption of refractory powder to mass consumption of nitrogen *µ* ratio vs. parameter *t_x_*/*t*_1_ (––) and rate ratio *w_x_*/*w*_1_ (- ) at a distance *x* from the nozzle cross-section.

**Figure 6 materials-14-02289-f006:**
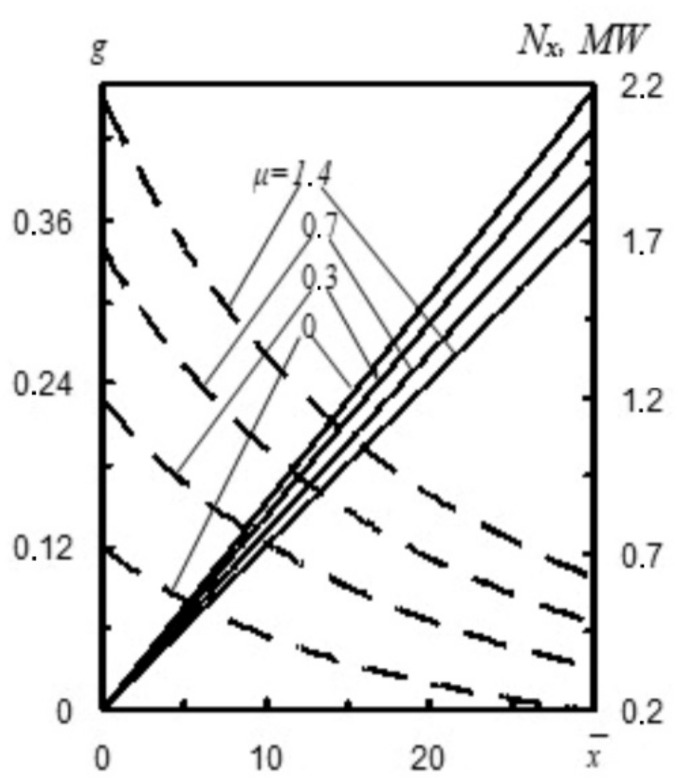
Mass *g* (––) and stream power *N_x_* (- ) vs. mass consumption of refractory powder to mass consumption of nitrogen *µ* ratio at a distance *x* from the nozzle cross section [17].

**Figure 7 materials-14-02289-f007:**
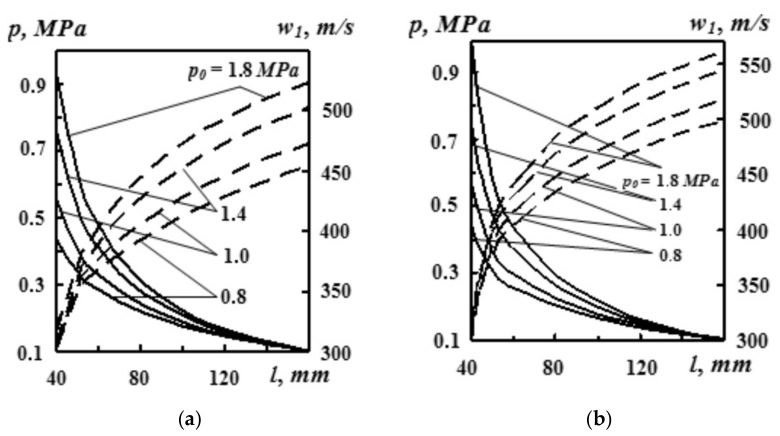
Pressure *p*_0_ vs. distribution of static pressure *p* (––) and gas flow rate *w*_1_ (---) at a distance l from the broadening part of the nozzle. Source data: (**a**) *μ* = 1.5 kg·kg^−1^ and (**b**) *μ* = 0.3 kg·kg^−1^ [8].

**Figure 8 materials-14-02289-f008:**
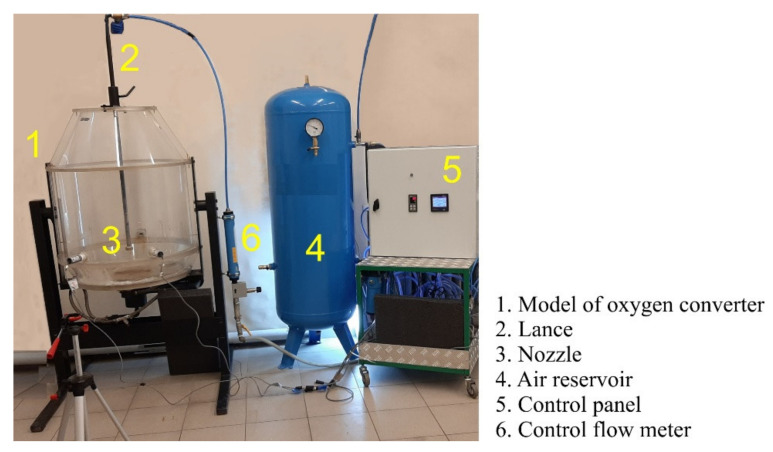
View of the physical model of oxygen converter.

**Figure 9 materials-14-02289-f009:**
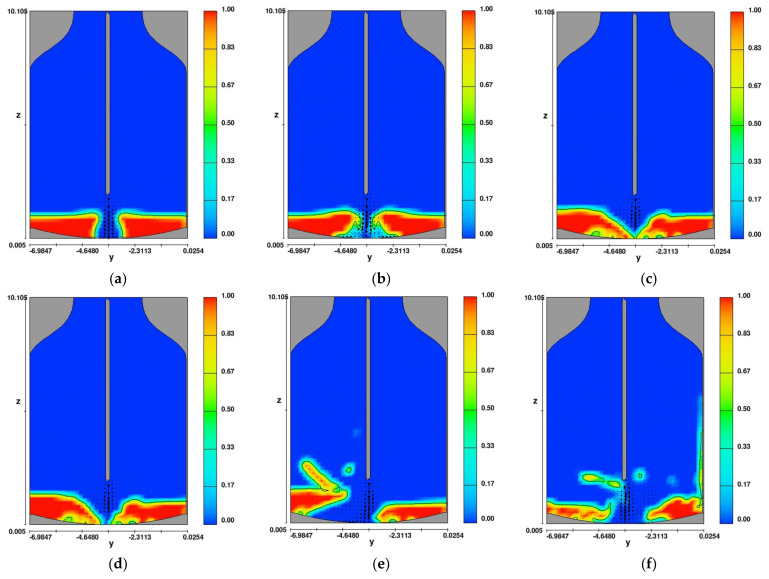
Distribution of liquid phase for variant 1 after time (**a**) 0.05 s, (**b**) 0.1 s, (**c**) 0.45 s, (**d**) 0.5 s, (**e**) 1.5 s, and (**f**) 2.5 s.

**Figure 10 materials-14-02289-f010:**
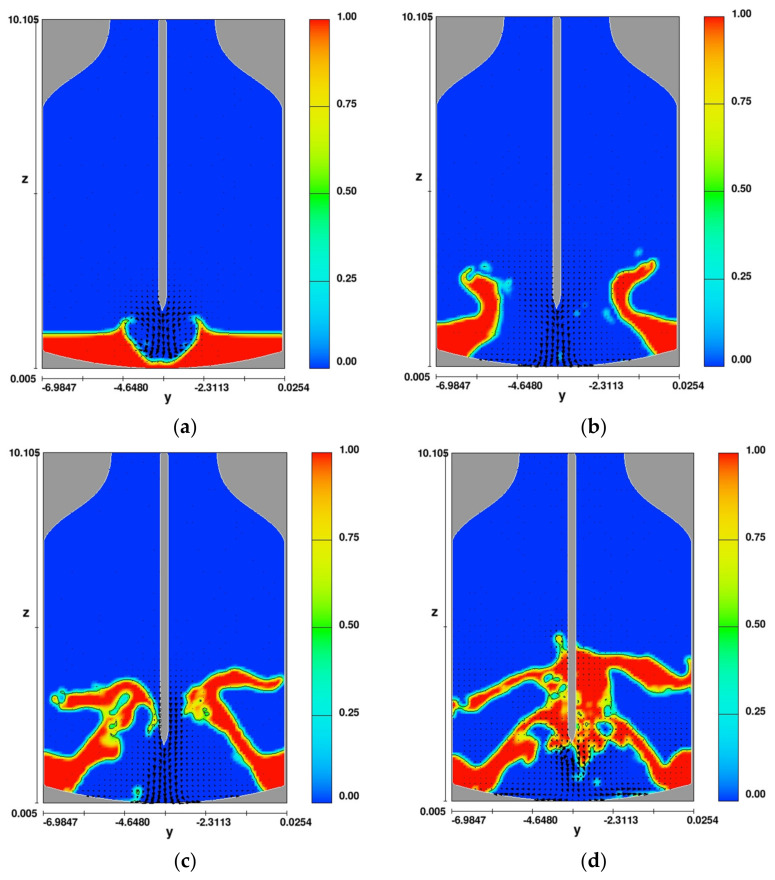
Distribution of phases in variant 1 for time (**a**) 0.05 s, (**b**) 0.45 s, (**c**) 0.65 s, and (**d**) 0.85 s.

**Figure 11 materials-14-02289-f011:**
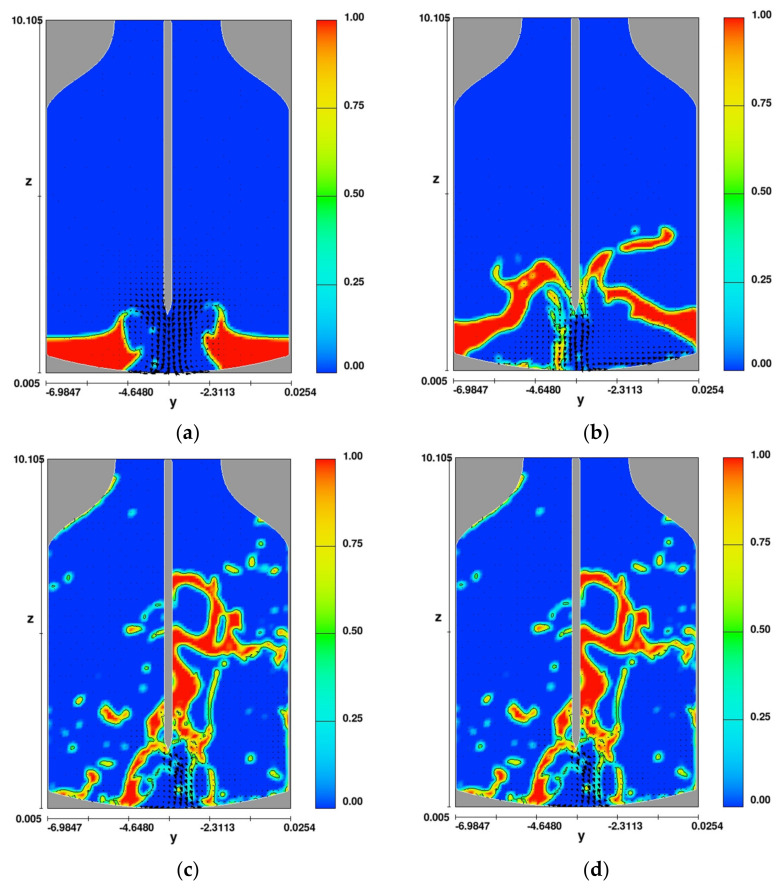
Distribution of phases for variant 1 for time (**a**) 0.1 s, (**b**) 0.45 s, (**c**) 0.1 s, and (**d**) 2.35 s.

**Figure 12 materials-14-02289-f012:**
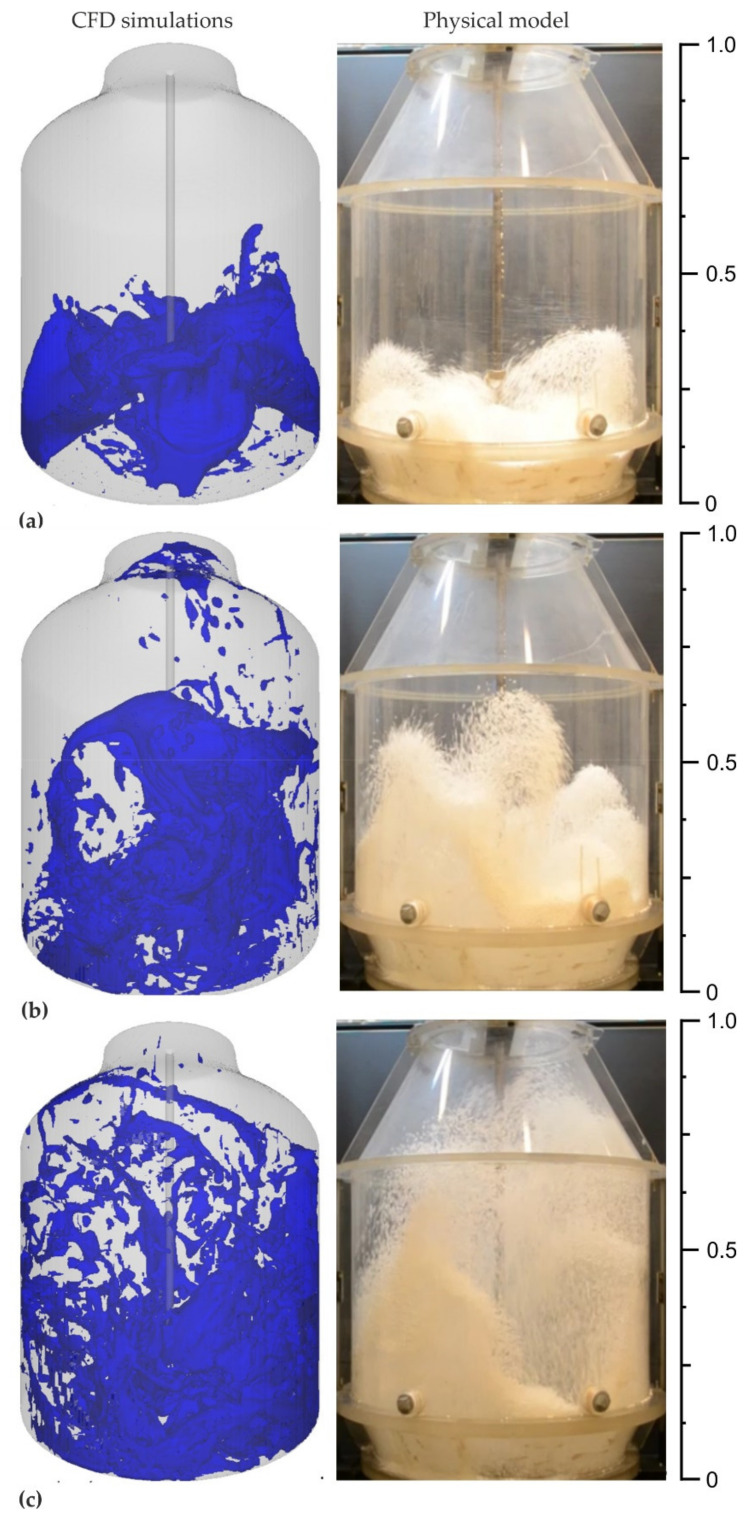
Verification of CFD simulations—physical model for time (**a**) 0.45 s, (**b**) 1.0 s, and (**c**) 2.35 s.

**Table 1 materials-14-02289-t001:** Parameters adopted for CFD calculations.

Parameter	Variant Designation		
	1	2	3
Height of the lance above the slag	1.5 m		
Initial molten slag depth	0.5 m		
Inside diameter of the lance	0.25 m	0.2 m	
Nitrogen flow through one nozzle	1.6 m^3^·s^−1^		
Diameter of nozzles	0.032 m		
Number of nozzles	1	4	5
Exit angle of nozzles	0 degrees (oriented directly at the liquid slag)	14 degrees	4–14 degrees; 0 degrees (oriented directly at the liquid slag)

**Table 2 materials-14-02289-t002:** Properties of fluids used for CFD calculations.

Parameter		Unit	Value
Slag	Density	kg·m^−3^	3000
	Viscosity	kg·m^−1^·s^−1^	0.07
Nitrogen	Density	kg·m^−3^	1.225
	Viscosity	kg·m^−1^·s^−1^	6 × 10^−5^

**Table 3 materials-14-02289-t003:** Experiment parameters for the physical modeling.

Parameter	Unit	Value	
		Industrial Scale (1:1)	Model Scale (1:10)
Height	m	10.1	1.01
Diameter	m	7.8	0.78
Diameter of nozzles	m	0.032	3.2 × 10^−3^
Number of nozzles	-	5	5
Inside diameter of the lance	m	0.2	0.02
Initial molten slag depth	m	0.5	0.05
Height of the lance above slag/granulate	m	1.5	0.15
Nitrogen/Airflow intensity	m^3^·s^−1^	8.0	1.7 × 10^−3^
Nitrogen/Air density	kg·m^−3^	1.225	1.204
Nitrogen/Air viscosity	kg·m^−1^·s^−1^	6 × 10^−5^	1.516 × 10^−5^
Slag/Granulate polystyrene density	kg·m^−3^	3000	27

## Data Availability

Data are contained within the article.

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
