# Peer review of "Physical and Numerical Modeling of the Slag Splashing Process"

_materials, 2021, doi:10.3390/ma14092289_

Round 1
Reviewer 1 Report
The presented article discusses the results of numerical and physical modeling of processes in the so-called "Slag Splashing" technology. This technology is known and widely used in BOF steel mills.
I have the following key commentary on the article:
- did the section on numerical and physical modeling reveal any findings that could be applied in practice? - at the end of the work it is declared only that there is a good agreement between the two methods of modeling
- the part devoted to physical modeling is greatly simplified - it does not consider the influence of viscosity and surface tension, which are two parameters that will have a significant effect on the "Slag Splashing" process
- Table 3 does not list the parameters for the operational converter, from which the degree of agreement in physical modeling should also follow.
I recommend reworking this part in accordance with the principles of physical modeling.
Author Response
Answers Reviewer 1
- did the section on numerical and physical modeling reveal any findings that could be applied in practice? - at the end of the work it is declared only that there is a good agreement between the two methods of modeling
The analyzed model creates bases for numerical modeling of the nozzle operation in a wide range of carrier gas flow, also when refractory powder is added (μ = 0 - 30 kg nitrogen / kg powder). The numerical calculations with the slag splashing program show that the use of heating of the carrier medium (nitrogen) is advantageous for the slag splashing effect as then an additional temperature effect takes place affecting the slag heating, and so, also the physicochemical parameters. The more durable is the effectiveness of pneumatic slag splashing on the hot lining of the oxygen converter, the higher is the power of the splashing gas flow rate. At the same time, simulation results show that the slag spattering affects the interaction of supersonic gas streams with liquid slag, where even a small amount of slag added to the stream significantly inhibits its velocity. It has also been observed that preheating carrier gas, e.g. by heat recovery from the process, lowers the cost of the operation as the nitrogen consumption is reduced without changing the stream power. With the applied calculation procedure the nozzle shape can be so selected as to ensure the optimum flow of the gas stream or gas-powder mixture for any gas pressure variant. Consequently, with the selected process parameters the power of the stream flowing into the slag can be increased for the same gas or gas-powder mixture consumption. Modeling with the program creates opportunity to further improve and develop the slag splashing technology by introducing changes in the nozzle design and using heat recovery from the process. Simulations with Flow 3 D show that the most effective slag distribution on the converter walls was obtained for the lance with a diameter of 0.2 m and 4 nozzles placed at an angle of 14° with an additional, centrally placed nozzle, perpendicular to the liquid slag surface. A similar design is currently in operation at the "АрселорМіттал Кривий Ріг" steel plant in Ukraine and at "NLMK" in Russia.
- the part devoted to physical modeling is greatly simplified - it does not consider the influence of viscosity and surface tension, which are two parameters that will have a significant effect on the "Slag Splashing" process
The numerical modeling presented in this paper lied in evaluating the effects of thermal and kinetic parameters as well as the nozzle design on slag spattering. The effect of physicochemical properties on slag viscosity and surface tension, which affect slag spattering and adhesion to the furnace lining as well as slag interaction with the gas stream, were not analyzed. The influence of slag physicochemical parameters have been studied theoretically by the authors, as published in the following papers:
- Kalisz, D. Investigation of the physicochemical properties of slag splashed on the lining of an oxygen converter / D. Kalisz, V.O. Sinelnikov, K. Kuglin // Refractories and Industrial Ceramics – 2018. – Vol. 55, № 5. – P. 463-468.
- Sinelnikov V.O. Modeling viscosity of converter slag / V.O. Sinelnikov, D. Kalisz // Archives of Foundry Engineering. – 2015. – Vol. 15, № 4 (Special Issue). – P. 119-124.
- Sinelnikov V.O. Influence the FeO content on slag viscosity at his spraying. Increase the life of the refractory lining / V.O. Sinelnikov, D. Kalisz // Glass and Ceramics. – 2016. – Vol. 73, issue 3 – 4. – P. 144-148.
- Table 3 does not list the parameters for the operational converter, from which the degree of agreement in physical modeling should also follow.
The Table 3 was completed
I recommend reworking this part in accordance with the principles of physical modeling.
Appropriate changes have been made in the chapter - 2.4. Physical Modeling
The attachment contains the text of the article with changes according to comments from All of Reviewers

Reviewer 2 Report
This manuscript studied the influence of technological factors on the process of slag splashing from the physical and numerical views. Several factors that affecting the slag splashing were highlighted such as flow parameters, pressure and temperature of the nitrogen stream, etc. Some comments should be addressed after reviewing this study:
1 The research gap (why did this research) was not well represented in the introduction, while it is more like a methodology illustration. A figure about the configuration of slag splashing may be better for readers to have a direct impression on this technique. In addition, for readers who are not in this specific field, it is hard for them to understand the novelty of this research.
2 A schematic diagram of the modelling strategy, or a diagram of the models would help the readers understand the modelling section.
3 The resolution of figure 2 (b) is much higher than that of figure 1 and figure 2 (a), which could be improved, otherwise it is like the ‘screenshot’ picture. Same issues in figure 3, 4 and 5.
4 In figure 6, annotations are desired to name different parts in the configuration. A picture like this without any annotations is definitely not suitable for scientific publications.
5 In verification stage 2, the indicators of the verification were not clearly presented, including the errors, error bars, etc. Images could not proceed precise comparison unless all the indicators are quantified.
Author Response
Answers Reviewer 2
- The research gap (why did this research) was not well represented in the introduction, while it is more like a methodology illustration. A figure about the configuration of slag splashing may be better for readers to have a direct impression on this technique. In addition, for readers who are not in this specific field, it is hard for them to understand the novelty of this research.
After the steel is melted in the oxygen converter and poured into the ladle, a certain amount of molten slag is left at the bottom of the furnace. This slag is modified with additives that increase its ability to adhere to the refractory lining of the converter. Then the lance is lowered and a stream of nitrogen or a nitrogen-MgO powder mixture is blown into the liquid slag, resulting in its splashing on the walls of the refractory lining. The high-pressure gas blowing is carried out for 2 to 4 minutes, the process results in the formation of a protective slag layer on the surface of the spent refractories, which provides better protection and maintenance of the refractory. The excess slag is then poured out of the converter to prevent clogging of the nozzles of the gas permeable fittings in the bottom of the converter (Figure 1).
The following technological and design parameters influence the thickness and distribution of the applied slag layer: stream characteristics (flow rate, lance height and angle, number and dimensions of nozzles), slag physicochemical properties (viscosity, density, etc.), system geometry.
(New figure in the text (attachment))
Figure 1. Scheme of slag splashing in the oxygen converter.
- A schematic diagram of the modelling strategy, or a diagram of the models would help the readers understand the modelling section.
(New figure in the text (attachment))
Figure 2. Scheme research plan
- The resolution of figure 2 (b) is much higher than that of figure 1 and figure 2 (a), which could be improved, otherwise it is like the ‘screenshot’ picture. Same issues in figure 3, 4 and 5.
Figures have been corrected
- In figure 6, annotations are desired to name different parts in the configuration. A picture like this without any annotations is definitely not suitable for scientific publications.
Description in Figure 6 has been supplemented
- In verification stage 2, the indicators of the verification were not clearly presented, including the errors, error bars, etc. Images could not proceed precise comparison unless all the indicators are quantified.
The text of the article has been supplemented
The attachment contains the text of the article with changes according to the comments from All of Reviewers

Round 2
Reviewer 1 Report
The problem of physical modeling of slag spattering during gas blowing is much more complicated than the authors state. The use of the Navier-Stokes equation to derive similarity criteria is also controversial. Their determination should be based on dimensional analysis, as this is a process in which slag droplets are formed from the liquid phase. Therefore, even such parameters as surface tension and viscosity will play a big role. In addition, Table 3 shows that the condition of similarity in the density of real slag and granular polystyrene was far from met, which may affect the results. It should be mentioned in the article that this is a very complex process with incomplete agreement of the relevant similarity criteria. Personally, I would rather call this chapter as "a simulation of the behavior of slag during gas blowing on a reduced water model".
Author Response
The problem of physical modeling of slag spattering during gas blowing is much more complicated than the authors state. The use of the Navier-Stokes equation to derive similarity criteria is also controversial. Their determination should be based on dimensional analysis, as this is a process in which slag droplets are formed from the liquid phase. Therefore, even such parameters as surface tension and viscosity will play a big role. In addition, Table 3 shows that the condition of similarity in the density of real slag and granular polystyrene was far from met, which may affect the results. It should be mentioned in the article that this is a very complex process with incomplete agreement of the relevant similarity criteria. Personally, I would rather call this chapter as "a simulation of the behavior of slag during gas blowing on a reduced water model".
The Reviewer's remarks are of course correct. For comprehensive research aimed at identifying the phenomena occurring during slag splashing in the oxygen converter as a result of dimensional analysis, similarity criteria such as Fr, Re, We, Eu should be taken into account. However, simultaneous fulfillment of these criteria is very difficult (the case of simultaneous fulfillment of Fr and Re). The presented research was simplified, taking into account the limited purpose of this research (visualization of the slag spattering mechanism). As the Reviewer rightly noted, the model used in the research is a model with incomplete compliance with the relevant criterion numbers.
In the article, changes in the text are marked (yellow).
Reviewer 2 Report
Comments have been addressed, can be accepted
Author Response
In the article, changes in the text are marked (yellow).